# Reliability of a Musculoskeletal Assessment for the Examination of Cervical Spine Pain and Injuries in Special Forces Combat Soldiers

**DOI:** 10.3390/sports12090255

**Published:** 2024-09-14

**Authors:** Timothy C. Sell, Ryan Zerega

**Affiliations:** Atrium Health Musculoskeletal Institute, Charlotte, NC 28207, USA; ryan.zerega@atriumhealth.org

**Keywords:** reliability, neck, cervical spine, range of motion, neck strength, injury prevention, intra-rater, inter-rater, neck range of motion

## Abstract

An assessment of protocol reliability is an essential step prior to human subject testing for injury prevention. The purpose of this study was to examine the inter-rater and intra-rater reliability of a portable cervical range of motion and isometric strength protocol designed for special forces combat soldiers who are at risk for cervical spine pain and injury due to exposure to head-supported mass. Eight individuals were tested three times to assess reliability, the standard error of the measurement (SEM), and the minimal detectable change across six range of motion measures and six strength measures of the cervical spine. One tester tested all participants twice for intra-rater reliability, and a second tester assessed the participants to examine inter-tester reliability. All reliability measures demonstrated good to excellent reliability (ICC = 0.70–0.96 (isometric strength); ICC = 0.85–0.94 (range of motion)). All SEM scores were 12% or lower for all reliability measures. The findings of this study demonstrate that the protocol developed for a longitudinal multi-site study is reliable and appropriate to implement for injury prevention in military personnel.

## 1. Introduction

Musculoskeletal injuries pose a significant concern regarding the health of military populations [1,2]. The problem continues to worsen [1,2], with cervical spine (c-spine) injuries and neck pain being particularly problematic [3,4]. Some populations, including special forces operators, helicopter pilots, and fighter pilots, are more susceptible to cervical spine injuries [5,6,7,8,9] than other populations as they perform missions that include parachute opening events and ground-based activities in demanding environments with significant head-supported mass (HSM) [10]. This head-supported mass can consist of a protective helmet, communication gear, and specialized vision technology weighing upwards of 3 kg [11]. Special forces operators such as Naval Special Warfare Sea, Air, and Land operators and United States Army Special Operations Forces who must perform in challenging environments with this HSM are at risk for both chronic and acute injury to the c-spine [10]. The impact of injuries on the health and readiness of military personnel justifies the need to determine the effects of HSM on the musculoskeletal characteristics of the cervical spine, the determination of modifiable risk factors for these injuries, and appropriate rehabilitation strategies and modalities [12,13,14,15,16]. In addition, it is critical to determine whether these injuries can be prevented and what interventions would effectively prevent them. The cervical spine’s strength and range of motion are two potential targets for injury prevention [17,18]. Previous research has demonstrated that pilots with longer exposure to HSM see changes in their cervical range of motion and strength and that pilots with cervical pain have a worse range of motion than pilots without cervical pain [19,20]. The first step in assessing these characteristics as part of research must be the development of a reliable protocol that can be implemented across multiple test sites and assessment days. 

Cervical range of motion and strength testing has been used previously in research utilizing multiple technologies in multiple populations, including military personnel [19,20,21,22,23]. For example, Palmieri and colleagues examined the reliability of cervical range of motion measurement in 20 healthy young adults utilizing inertial sensors (a triaxial accelerometer and a triaxial gyroscope) [24]. Using this technology, they determined that their protocol was reliable for cervical range of motion [24]. Nagai et al. have examined the reliability of measurement of cervical range of motion and strength and the standard error of the measurement (SEM) as part of research investigating the effects of neck pain on strength, flexibility, and posture in helicopter pilots [19,20]. For their study, they utilized the CROM 3 (Performance Attainment Associates, Lindstrom, MN, USA) and a hand-held dynamometer (HHD) (Lafayette Instruments, Lafayette, IN, USA). They demonstrated good to excellent reliability and SEM across six movements for range of motion and strength. A recent systematic review examined the reliability and validity of clinical tests for measuring the strength of cervical muscles [25]. A total of 26 studies met their inclusion/exclusion criteria. Overall, the results indicated that HHD testing for cervical strength was acceptable, but it was clear that different research groups have utilized different technologies and protocols for their research needs. Furthermore, some groups have examined different forms of reliability (intra-rater vs inter-rater reliability), with the results varying across research studies. This variability and difference in previous protocols demonstrate the need to develop protocol-specific and rater-specific reliability measures for each injury prevention study and those testers involved. In addition, it is vital to establish reliability with the specific measurement devices employed in proposed studies (e.g., a specific hand-held dynamometer). 

The ability to identify individuals at risk for cervical spine injury and pain is an essential step in the prevention of these injuries. Before this step, it is necessary to develop reliable protocols for all assessments in anticipation of longitudinal testing and multiple testers. Reliability should also be specific to the instrumentation and protocols. Therefore, this project aimed to develop reliable (intra-rater and inter-rater) test protocols to assess cervical range of motion and strength in a healthy population. Two specific aims were developed: (1) to examine the intra-rater reliability and (2) to examine the inter-rater reliability of cervical spine active range of motion and cervical spine strength testing with a hand-held dynamometer. As part of this study, we also wanted to calculate the minimal detectable change for all measures performed. We hypothesized that all measurements would demonstrate good reliability (intra-rater and inter-rater) and a low standard of error of the measurement. The results of this study will inform testing protocols for cervical ROM and strength testing in military populations exposed to significant HSM. 

## 2. Materials and Methods

### 2.1. Participants

Participants were healthy, physically active individuals recruited via email and word of mouth. Eight individuals (four females and four males) participated in the study (age = 25.8 ± 6.2 years; height = 173.5 ± 8.9 cm; weight = 72.5 ± 15.8 kg). All of these participants were healthy and physically active with no history of cervical spine surgery, no recent (within one year) history of cervical spine injury, no disorders that could affect equilibrium or neuromuscular control, and no history of head injury in the past three months. All participants voluntarily consented to the study and signed an informed consent form approved by the Wake Forest University School of Medicine Institutional Review Board.

### 2.2. Instrumentation

The CROM 3 (Performance Attainment Associates, Lindstrom, MN, USA) was used to measure the active range of motion. The CROM 3 is an analog device with two bubble goniometers and a magnetic compass (see Figure 1). Isometric cervical strength testing was performed with a hand-held dynamometer (HHD) (Hoggan Scientific, Salt Lake City, UT) (see Figure 2).

### 2.3. Protocol Development

Before any testing for this study, a standard operating procedure (SOP) was developed for cervical active range of motion (CAROM) and cervical strength (isometric) testing. This SOP included cervical flexion, extension, rotation (bilaterally), and lateral flexion (bilaterally) for CAROM and strength. The SOP was based on several previous studies that the current investigators were part of, including testing helicopter pilots and Naval Special Warfare crewmen [19,20,21,26]. Developing the SOP for this study began with carefully reviewing the original procedures employed in these previous studies. The instrumentation, participant positioning, and test procedures were reviewed. Following this review, three different test and review sessions were performed. In the first test session, investigators TS and RZ performed the CAROM measurements on four individuals. Following this test session, the procedures were discussed with the individual participants for feedback on verbal instructions and within the investigator team, and then the SOP was reviewed. Modifications to the SOP included ensuring consistency with the previous protocol and consistency between testers and participants. Some examples of these modifications included the following:Specific verbal cues during data collection such as “Ready, Set, Go, 3, 2, 1, relax”;Positioning of the participant during testing, such as lowering the examination table to its lowest position to maximize the tester’s mechanical advantage to resist cervical flexion and extension;Guidelines for placement of the HHD pad during testing;The decision to perform CAROM before strength testing to ensure that strength testing would not impact CAROM testing due to concerns about the potential for pain provocation;Verbal cues for the participant during CAROM to ensure the proper start position before each CAROM repetition/trial.

These procedures were then implemented during a test session for strength testing with these same subjects. Finally, all participants underwent testing of both CAROM and strength testing by TS and RZ during one single test session. 

### 2.4. Procedures

Participants in the study reported for one research session that lasted approximately 90 min. Each participant was tested three times during this research session. The first two tests were used for inter-rater reliability, and the order was systematically randomized so that half of the participants were tested by TS first and the other half were tested by RZ first. The third test session was included for the intra-rater reliability objective and was only performed by TS. Participants sat quietly in a chair for 10 min between test sessions to eliminate any potential issues with fatigue [27]. Testing began with CAROM and ended with strength testing. 

The CAROM measurements included cervical flexion, extension, lateral flexion (right and left), and rotation (right and left). Pillows were placed under both arms, on top of the armrests, to relax the shoulders. All CAROM tests began with the subjects’ heads in the Frankfort plane [28]. Subjects were instructed to move their head in the direction of the tested movement as far as possible until an uncomfortable stretch or pressure was felt. Testing included verbal cues such as “sit up tall”, “drop your ear to your shoulder”, and “rotate your head without moving your shoulders” to ensure movements were tested properly. The test order was cervical flexion/extension, right and left lateral flexion, and right and left rotation. Two practice trials were performed before the three test trials for data analysis. A fourth test trial was collected if any of the first three test trials was not within three degrees of the median value. A fifth test trial was collected if two or more trials were not within three degrees of the median value. A maximum of five test trials were performed. The absolute angle (degrees) relative to the subject’s neutral position (recorded before each movement) was recorded and averaged across the three test trials for statistical analysis. When more than three test trials were performed, the three test trials with the smallest standard deviation were used and averaged for statistical analysis.

Isometric strength was assessed for the same movements included in the CAROM testing. For the cervical extension strength testing, subjects were positioned prone on the treatment table with their face placed on a prone pillow, arms hanging over the sides of the table, and a pillow under the lower legs for comfort. The HHD stirrup was placed horizontally immediately over the occiput in the vertical midline of the head. For cervical flexion, subjects were positioned supine on a treatment table with their feet hip-width apart, a pillow under their knees, and their arms off the table in a “W” position. Before the flexion movement, subjects were asked to raise their heads off the table to a 45° angle, and the HHD stirrup was positioned across the forehead with the bottom edge slightly above the eyebrow line. The same position was used to assess lateral flexion and rotation, except that the subject’s hands were placed on the subject’s abdomen. In some cases, as necessary, the subject’s head rested on a one-inch-thick hardback book to maintain a neutral plumb-line alignment. A second researcher was positioned to stabilize the opposite side of the subject for both the lateral flexion and rotation testing. For the lateral flexion movement, the HHD stirrup was placed just above the ear on the lateral side of the head horizontally, with the second researcher stabilizing the shoulder opposite the testing direction. The HHD stirrup was placed horizontally over the subject’s frontal bone at the temporal line for rotational testing. The HHD stirrup was positioned perpendicular to the contour of the subject’s head before the flexion, lateral flexion, and rotation movements. Appropriate verbal cueing, similar to that described in the procedures for CAROM procedures, was also used. Before each movement was assessed, a warm-up procedure included one 50% maximum voluntary effort (MVE) trial followed by one 100% MVE warm-up trial. The trials to be collected for statistical analysis were at 100% MVE. Three trials, each lasting five seconds, were collected for each movement. As in CAROM testing, a fourth test trial was collected if any of the first three test trials was not within ten percent of the median value. A fifth test trial was collected if two or more trials were not within ten percent of the median. A maximum of five test trials were performed. 

### 2.5. Data Analysis

Statistical analysis began with calculating means and standard deviations for all variables. Intra-rater reliability was tested using an intraclass correlation coefficient (ICC) calculation (2,1 model for two-way random, single measure), and precision was determined by calculating the standard error of measurement (SEM) [29]. Similarly, inter-rater reliability was tested using an intraclass correlation coefficient (ICC) calculation (2,1 model for two-way random, single measure), and precision was determined by calculating the standard error of measurement (SEM) [29]. The following scale was used to interpret the ICC values: below 0.69 = poor, 0.70 to 0.79 = fair, 0.80 to 0.89 = good, and 0.90 to 1.00 = excellent [30,31]. The minimal detectable change (MDC) was calculated for each reliability measure (inter-rater and intra-rater). An alpha level of 0.05 was set a priori to determine the significance of all statistical analyses.

## 3. Results

The means and standard deviations for strength and CAROM values for each tester and test session are presented in Table 1. 

The ICC, SEM, and MDC calculations for both the inter-rater and intra-rater reliability are presented in Table 2. All the ICC values of intra-rater reliability were 0.90 or higher for the strength testing and 0.81 or higher for the CAROM testing. The inter-rater reliability for CAROM was generally higher than the intra-rater reliability, with all ICC values being 0.85 or greater. The inter-rater reliability for strength testing was lower than the intra-rater reliability, with the lowest value observed at 0.70. The rest of the inter-rater reliability scores were 0.83 or higher. 

## 4. Discussion

This study aimed to examine the intra-rater and inter-rater reliability of cervical spine range of motion and strength testing for a protocol that will be implemented to explore the effects of head-supported mass exposure in military personnel. As implemented in this study, the protocol resulted in good to excellent reliability for all the intra-rater and inter-rater reliability tests, excluding the inter-rater reliability of right lateral flexion strength testing. In addition, all assessments had a low standard of error of the measurement. Overall, these results supported our hypotheses that the protocol implemented in the current study would demonstrate good to excellent reliability and a low SEM. As such, the protocol implemented in this study is appropriate for further use in cervical ROM and strength testing as part of injury epidemiological and prospective risk factor analysis studies.

The results of this study are similar to those of previous studies utilizing similar protocols and equipment. The closest protocol to the one tested in the current study was employed in a study of Army helicopter pilots, although they utilized a different hand-held dynamometer than the one in the current study and did not test inter-rater reliability [19]. Nagai et al. utilized a hand-held dynamometer and the same active range of motion testing equipment. Their results regarding intra-rater (ICC 2,1) reliability for strength ranged from 0.79 to 0.97, which was similar to the range in the current study (0.90 to 0.97), and the corresponding results for active range of motion ranged from 0.53 to 0.98, which, at least for two of the measures, was also similar to the values in the current study (0.85 and 0.94). They did not assess inter-rater reliability. Palmieri et al. utilized inertial technology, including a triaxial accelerometer, magnetometer, and gyroscope, to evaluate a cervical range of motion test [24]. Their reliability measures (inter-tester) ranged from 0.85 to 0.98, similar to the current study’s inter-tester reliability (0.81 and 0.92). McBride et al. utilized the VALD ForceFrame (Newstead, Australia) to assess the inter-tester and intra-tester reliability of strength testing [32]. This portable system includes multiple load cell sensors fixed to a frame. They assessed lateral flexion, extension, and flexion in a quadrupedal position but did not assess rotation. Their inter-rater reliability measures (0.96 to 0.97) were similar in quality to the current study (good to excellent). They also evaluated intra-rater reliability but utilized the first two trials rather than two different sessions. These measures were similar between the two studies. 

Although most reliability numbers were demonstrated to be good to excellent, one measure, inter-rater reliability for right lateral flexion strength (ICC (2,1) = 0.70), was at the low end of the range for “good” reliability and was much lower than all of the other strength reliability measures (ICC (2,1) = 0.83–0.96) [30,31]. It was also low compared to the intra-rater reliability for the same movement (ICC (2,1) = 0.95). Statistically, this demonstrates, for this particular movement, that there were differences between testers. A qualitative examination of the participant data after completing the statistical analysis revealed that three of the eight participants appeared to have significant differences between the two testers. From these data, it is still challenging to determine why there were differences in these three comparisons, given the good to excellent results for all of the other measures, both strength and CAROM. More importantly, these results prompted the testers to discuss the potential reasons for these differences and to review the protocol in detail to ensure consistency in testing. It also serves as a warning for future research utilizing different testers and emphasizes the importance of practice, familiarity, and protocol review.

The results of the SEM and MDC calculations can be challenging to assess outside the context of the current study. In general, the SEM measures are better (lower) for intra-rater reliability than for inter-rater reliability, which is consistent with the ICC values as expected since they were also lower on average for intra-rater reliability. Typically, these values are less than 10% of the mean values for the subject testing in inter-rater testing and around 12% of the mean values in intra-rater testing. For reference, a perfect SEM would be 0.0. 

There are two important limitations to be considered for the current study. The first is that all testing was performed in a single day. Individuals underwent testing of all variables three different times. There is the possibility of fatigue, a warm-up effect, and familiarity with testing. To reduce the possibility of fatigue due to strength testing, each participant sat quietly in a chair for ten minutes between each test session. We attempted to reduce some of these concerns by systematically varying the test order and providing consistent instructions and warm-up procedures. This could affect the statistical analyses, although it is unclear whether it would negatively or positively impact the outcomes. Another potential limitation is the number of subjects included in the study (eight). The initial recruitment effort resulted in the enrollment of eight participants. We decided to run the proposed statistical analyses following the testing of these participants. With the exclusion of one result as outlined above, we were satisfied with the results. Individuals should take caution in interpreting these results and using these procedures, given the low number of subjects included in the study. We decided to proceed with testing of human subjects for the study for which this reliability study was necessary. We do not know what the results would be with further testing. Still, we would suggest that additional testing would have only improved the procedures and testing proficiency, likely resulting in better results. 

## 5. Conclusions

The testing of neck strength and range of motion is an important part of identifying, predicting, and preventing cervical spine injuries and pain in military personnel, especially those who must wear significant head-supported mass. To be successful, it is necessary to develop methodological protocols specific to the equipment and research design—in this instance, prospective and longitudinal studies that involve multiple test locations. This may require the use of portable equipment and multiple testers. The current study demonstrates, at a minimum, good intra-rater and inter-rater reliability for all strength and range of motion testing with portable equipment.

## Figures and Tables

**Figure 1 sports-12-00255-f001:**
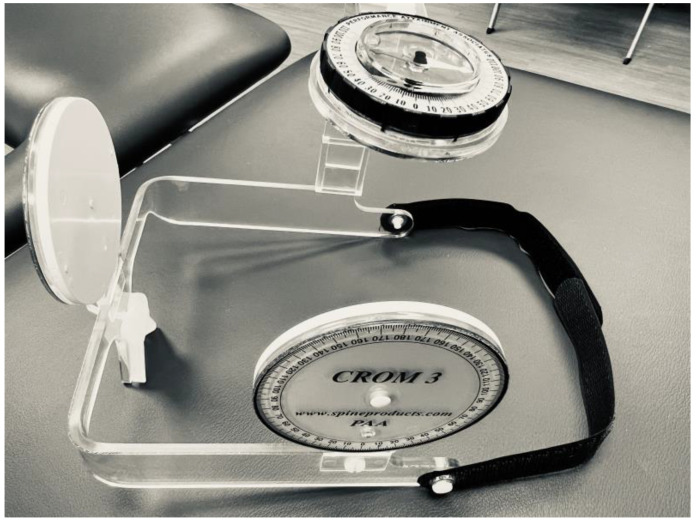
Device used for active range of motion testing.

**Figure 2 sports-12-00255-f002:**
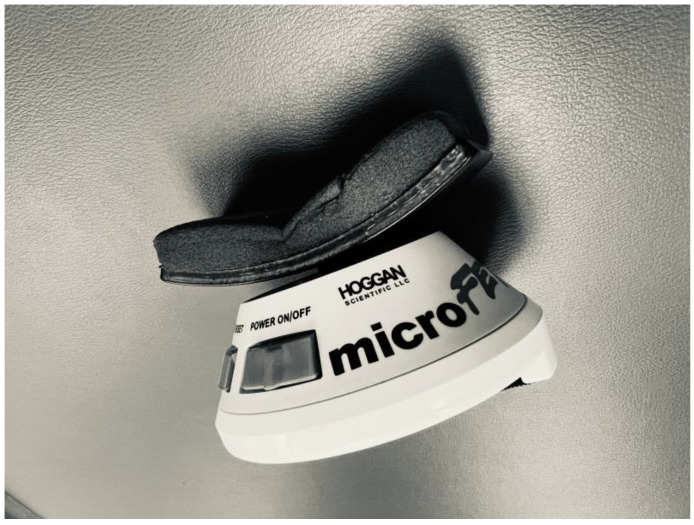
Hand-held dynamometer used for strength testing.

**Table 1 sports-12-00255-t001:** Means and standard deviations for each tester and each test session.

		Session 1				Session 2	
		Tester 1		Tester 2		Tester 2	
		Mean	Standard Deviation	Mean	Standard Deviation	Mean	Standard Deviation
Strength	Extension (Newtons)	164.6	54.6	185.3	52.2	202.2	60.3
	Flexion (Newtons)	79.4	32.1	81.0	31.2	78.5	30.2
	Right Rotation (Newtons)	116.4	37.2	122.5	26.9	133.1	30.1
	Left Rotation (Newtons)	118.5	44.0	124.0	31.4	124.9	29.6
	Right Lateral Flexion (Newtons)	117.5	40.6	136.9	32.7	137.0	38.1
	Left Lateral Flexion (Newtons)	124.5	47.2	128.2	32.9	131.2	31.6
Range of Motion	Extension (Degrees)	72.1	17.5	72.8	15.4	72.0	12.8
	Flexion (Degrees)	63.5	11.7	63.8	11.1	64.2	10.9
	Right Rotation (Degrees)	69.3	7.9	66.3	12.8	66.2	11.1
	Left Rotation (Degrees)	71.6	8.5	68.3	11.6	70.7	9.6
	Right Lateral Flexion (Degrees)	46.0	8.4	48.0	9.2	47.6	7.0
	Left Lateral Flexion (Degrees)	47.7	8.6	49.8	9.8	49.3	9.3

**Table 2 sports-12-00255-t002:** Intraclass correlation coefficient, standard error of the measurement, and minimal detectable change values.

		Inter-Rater			Intra-Rater		
		ICC	SEM	MDC	ICC	SEM	MDC
Strength	Extension (Newtons)	0.83	21.9	30.9	0.93	14.4	20.4
	Flexion (Newtons)	0.96	6.1	8.7	0.95	6.7	9.5
	Right Rotation (Newtons)	0.83	13.1	18.5	0.90	8.8	12.5
	Left Rotation (Newtons)	0.89	12.3	17.4	0.97	5.3	7.4
	Right Lateral Flexion (Newtons)	0.70	20.4	28.9	0.95	7.7	10.9
	Left Lateral Flexion (Newtons)	0.89	13.0	18.4	0.95	6.7	9.5
Range of Motion	Extension (Degrees)	0.94	3.4	4.8	0.92	4.4	6.2
	Flexion (Degrees)	0.94	2.5	3.5	0.87	4.0	5.6
	Right Rotation (Degrees)	0.90	3.6	5.1	0.85	4.0	5.7
	Left Rotation (Degrees)	0.93	2.8	4.0	0.85	3.9	5.5
	Right Lateral Flexion (Degrees)	0.85	3.0	4.3	0.81	3.7	5.2
	Left Lateral Flexion (Degrees)	0.88	3.3	4.6	0.86	3.4	4.8

ICC = intraclass correlation coefficient; SEM = standard error of the measurement; MDC = minimal detectable change.

## Data Availability

We can make the data available but will need to explore secure/safe means of doing so without compromising any other subject or protected health information.

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
