# Peer review of "Reliability of a Musculoskeletal Assessment for the Examination of Cervical Spine Pain and Injuries in Special Forces Combat Soldiers"

_sports, 2024, doi:10.3390/sports12090255_

Round 1

Reviewer 1 Report

Comments and Suggestions for Authors

The authors are to be commended for their efforts to research reliability of cervical ROM and strength testing. The paper is generally well written and presented, but a little more detail and some changes to the document are required to improve it.

In the introduction the authors suggest a link between cervical ROM/strength and pain. But is the link causal or just association. Are the changes in ROM etc due to pain or the cause of the pain. This is pretty important as it is the crux of the study.

There is a bit of jargon that might not be familiar to the reader. Eg Force health

Abbreviations are not consistent. Please check the paper carefully. Particularly HSM, SEM 

If the protocol for ROM and strength testing has been evaluated previously why does this study need to repeat it?

There is unnecessary repletion. Eg line 65. More could be given to the need for this study and why there is a gap in the literature. 

Did you follow reliability study guidelines? Eg GRRAS guidelines

If this study is the basis for other studies on military personnel then it would be important to evaluate reliability in such subjects. How were subjects chosen? Volunteers? 

I would have thought metric reporting would be standard. 

What experience did the examiners have?

Were measures blind?

Describe the CROM 3. Is it an electrogoniometer? Analogue? A picture would be useful.

How did you determine sample size. It seems very small. Also the method of discarding trials outside 10% of the median value seems to artificially inflate reliability. What is the justification for this. 

3 decimal places is unnecessary. 

Line 208. What is meant by proper?

The conclusion overreaches the study findings. You need to completely re-write this and focus on what you found.

Line 217. You don’t need to mention that one author was in another study. 

How does your protocol for ROM and strength testing differ to the helicopter study. It is not clear why this study needed to be repeated if the same protocol was used. Or was there a reason to change?

Line 240 to 246 is a concern. It appears to read that the authors checked the data during testing and changed the protocol midway through as there was inconsistency? Is that correct? Its OK to do piloting, but not to vary the procedure during the main study.  

Reviewer 2 Report

Comments and Suggestions for Authors

General Comments:

This study presents a well-designed and executed investigation into the reliability of cervical spine range of motion and strength testing protocols. The research is particularly relevant given the high incidence of cervical spine injuries among military personnel, especially those exposed to head-supported mass. The study addresses a critical issue in military health, focusing on a population (special forces combat soldiers) that is especially susceptible to cervical spine injuries. However, I have several suggestions to further enhance the overall quality of the manuscript:

1.     Line 83: The small sample size (n=8) is a limitation of the study. While the authors acknowledge this in the limitations section, it would be beneficial to include a power analysis or provide justification for this sample size. Consider discussing how the small sample size might impact the generalizability of the results.

2.     Line 97: The authors mention that the SOP was based on previous studies. It would be helpful to provide more details on how this SOP was developed and refined, particularly any changes made based on the initial practice sessions. If possible, consider adding photographs to illustrate the measurement context, aiding reader comprehension.

3.     Line 176: Please clarify what is meant by the "(2,1 model)."

4.     Line 184: The results section would benefit from a more narrative structure. Currently, it mainly restates data from the tables. Consider discussing patterns or trends in the data more explicitly.

5.     Line 234: When reporting "Although most reliability numbers were demonstrated to be good to excellent, one measure, inter-rater reliability for right lateral flexion strength (ICC (2,1) = 0.696), was at the low end of the range for ‘good’ reliability and was much lower compared to all of the other strength reliability measures (ICC (2,1) = 0.828 – 0.960)," I believe that an ICC of 0.696 may not be sufficient for this type of measurement. Please provide evidence, such as references, to support this conclusion.

6.     Line 256: The authors note that conducting all tests in a single day could introduce fatigue or learning effects. It would be valuable to discuss in more detail how these potential confounding factors were mitigated and how they might have affected the results in the Methods section.

Minor/Specific Comments:

1.     Although SI units are more commonly used than BS units, it may be challenging to convert the entire article to SI units.

2.     Line 84: The "plus/minus" sign appears to be missing.

3.     Tables 1 and 2: The tables are presented as images, but the quality is poor. Please correct this.

4.  Ensure that all references follow the same format. References are incorrectly cited in the text, often immediately after a sentence rather than after the period.

Round 2

Reviewer 1 Report

Comments and Suggestions for Authors

The authors have made the recommended changes to the manuscript. While there is an issue with the sample size, this has been recognised as a weakness. 

Reviewer 2 Report

Comments and Suggestions for Authors

I truly appreciate the authors for their diligent efforts in revising the manuscript. Their careful and detailed revisions are evident. I have thoroughly reviewed the authors' responses to my previous questions and concerns, and I found that all of them have been well addressed, whether through revision or clarification. I am pleased to say that I have no further comments on the revised manuscript.